# Energy-Efficient Elderly Fall Detection System Based on Power Reduction and Wireless Power Transfer

**DOI:** 10.3390/s19204452

**Published:** 2019-10-14

**Authors:** Sadik Kamel Gharghan, Saif Saad Fakhrulddin, Ali Al-Naji, Javaan Chahl

**Affiliations:** 1Department of Medical Instrumentation Techniques Engineering, Electrical Engineering Technical College, Middle Technical University, Baghdad 10010, Iraq; saif_hm87@yahoo.com (S.S.F.); ali_al_naji@mtu.edu.iq (A.A.-N.); 2College of Dentistry, University of Mosul, Mosul 41002, Iraq; 3School of Engineering, University of South Australia, Mawson Lakes, SA 5095, Australia; Javaan.Chahl@unisa.edu.au; 4Joint and Operations Analysis Division, Defence Science and Technology Group, Melbourne, VIC 3207, Australia

**Keywords:** accelerometer, battery life, data-event algorithm, fall detection, heartbeat, power saving, GPS, GSM, WPT

## Abstract

Elderly fall detection systems based on wireless body area sensor networks (WBSNs) have increased significantly in medical contexts. The power consumption of such systems is a critical issue influencing the overall practicality of the WBSN. Reducing the power consumption of these networks while maintaining acceptable performance poses a challenge. Several power reduction techniques can be employed to tackle this issue. A human vital signs monitoring system (HVSMS) has been proposed here to measure vital parameters of the elderly, including heart rate and fall detection based on heartbeat and accelerometer sensors, respectively. In addition, the location of elderly people can be determined based on Global Positioning System (GPS) and transmitted with their vital parameters to emergency medical centers (EMCs) via the Global System for Mobile Communications (GSM) network. In this paper, the power consumption of the proposed HVSMS was minimized by merging a data-event (DE) algorithm and an energy-harvesting-technique-based wireless power transfer (WPT). The DE algorithm improved HVSMS power consumption, utilizing the duty cycle of the sleep/wake mode. The WPT successfully charged the HVSMS battery. The results demonstrated that the proposed DE algorithm reduced the current consumption of the HVSMS to 9.35 mA compared to traditional operation at 85.85 mA. Thus, an 89% power saving was achieved based on the DE algorithm and the battery life was extended to 30 days instead of 3 days (traditional operation). In addition, the WPT was able to charge the HVSMS batteries once every 30 days for 10 h, thus eliminating existing restrictions involving the use of wire charging methods. The results indicate that the HVSMS current consumption outperformed existing solutions from previous studies.

## 1. Introduction

Falls of elderly people are common in houses, hospitals, and health care centers, with 30 percent of falls leading to harm or injury. Nineteen percent of falls happen during movement from one location to another, and 84 percent occur in hospital rooms. Furthermore, the areas near beds and chairs are the locations of the majority of falls [1]. Most elderly people who fall during daily activities require special care, which limits their movement. The falls generally occur during elderly movement. Monitoring the vital signs of patients or elderly people, such as fall detection during daily activities using wireless body area sensor networks (WBSNs) after they are discharged from hospital, is important for achieving positive patient outcomes. This approach assists doctors in terms of monitoring a patient’s health status and providing first aid to them in critical situations [2]. Previous works have considered several proposed systems or wearable devices for monitoring patients’ vital parameters, such as heart rate [3], acceleration [4], SpO_2_ [5], and ECG [6]. Some of the related works adopted a GPS module to send the geographical location of the patient or elderly person to a central station monitored by the doctor [7]. In addition, these studies used various types of wireless technology such as WiFi [8], Bluetooth [9], ZigBee [10], and GSM [11] for transmitting patient parameters to the doctor for monitoring purposes and alerting them when critical situations occur.

One of the most significant challenges associated with designing wearable health monitoring devices is the power consumed by the components of these devices. When these components consume high amounts of power, battery lifetime decreases and the system becomes inefficient. Some past studies [12,13,14,15,16] did not consider approaches for solving this power consumption problem when designing patient monitoring systems or devices. In addition, these devices exhausted a considerable amount of current (~100–500 mA) because they continually monitored patients’ vital signs during daily activities. However, other research works have adopted power reduction techniques such as sleep/wake methods [17,18,19,20,21], data-driven algorithms [10], transmission power control (TPC) algorithms [22], modification of transmitted data [23], sampling and transmission rates [24], low-power idle mode [25], adaptive TPC [26], data rate control [27], routing protocol [28], and duty cycle [29] to reduce the power consumption of WBSNs for different applications such as fall detection, human activity monitoring, motion detection, vital signs monitoring, and localization. 

In WBSNs, most of the power consumption has been found to be dissipated during the transmission and reception process [30,31,32,33,34], specifically GSM. In recent works, radio components [35] such as GSM and GPS modules and microcontrollers have been considered power-hungry modules relative to the heart rate and accelerometer sensors, which consume a small amount of power. Reducing the power consumption of these components while collecting accurate measurements, obtaining geolocation information, and transmitting data to a remote location poses challenges. In this work, a human vital sign monitoring system (HVSMS) was proposed and prototyped for measuring vital parameters of the elderly, such as heart rate and acceleration, using two bio-sensors: an accelerometer and a heartbeat monitor. In addition, the upper arm was used as the location for attaching the HVSMS because the outer skin of the body in this location is very smooth and has high conductivity for the green LED of the photoplethysmography (PPG) sensor. The proposed system consisted of an accelerometer and a heartbeat sensor, a microcontroller based on the ATmega328p, a GPS module, a GSM module, and a 3.7 V/8400 mAh Li-ion rechargeable battery. Moreover, wireless power transfer (WPT) based on magnetic resonator coupling (MRC) was adopted for recharging the batteries of the proposed system to dispense with the need for charging by an electrical connection. In addition, a power reduction algorithm (i.e., data-event algorithm) was proposed to significantly limit the power consumed and increase the battery life of the HVSMS. The main operation of the proposed power reduction algorithm mainly depends on the active and sleep modes of the main components of the HVSMS (i.e., GSM and GPS). 

The DE algorithm was written inside the microcontroller. It leads to the GPS and GSM modules being turned off (i.e., zero power consumption) when the HVSMS in the patient monitoring system indicates that this switch is appropriate. In contrast, the DE algorithm places the GPS and GSM module in active mode and the biosensors in sleep mode when a patient fall occurs and the heart rate is normal or abnormal. Accordingly, in the last case, an alert message is sent from the HVSMS to the doctor in the emergency medical center (EMC) based on the GSM network. The microcontroller also sends three text messages to the EMC via GSM, and includes patient heart rate status and the geographical location of the patient fall (determined by GPS). Falls among elderly persons aged 65 years and older have gradually increased in recent years, according to the American Center for Disease Control and Prevention (CDC). More than 1 million elderly people fall and are treated in emergency departments due to a fall that causes a head injury or hip fracture each year in the USA. The measurement accuracy of elderly fall detection was experimentally tested for different ages (23 to 66 years), as presented in our previous work [36].

The contributions of the current work can be summarized as follows:A new prototype HVSMS was designed and implemented to monitor heart rate and detect falls of a patient or an elderly person, with a long communication range using the GSM network.HVSMS power consumption was significantly improved based on the DE algorithm, which utilizes the duty cycle of sleep/wake modes.The HVSMS was successfully charged based on an energy harvesting technique using WPT.HVSMS power consumption and battery life were enhanced by merging the DE algorithm and WPT.HVSMS power consumption was compared with relevant literature in terms of current consumption to validate its performance.

## 2. Related Works

A substantial challenge when designing wearable health monitoring devices is the current consumed by the components of these devices. The proposed vital signs monitoring system (VSMS) consumed a high current that led to a decrease in the life of the battery and decreased monitoring time. It was shown in References [13,14,15] that the authors did not consider any methods for decreasing current consumption when designing the devices. As a result of not including energy efficiency in the scope of these studies, these devices consumed high amounts of current (approximately 100–550 mA). Other works have, however, adopted algorithms for reducing power consumption, for example by including a sleep and wake scheme, a sampling strategy, a standby mode, and duty cycle control. These studies have had success in decreasing the consumed current and increasing battery lifetime. Authors in related works [17,19,20,21,24,37] have proposed wearable systems for health monitoring based on wireless body sensor networks (WBSNs). These studies adopted a sleep and wake algorithm to decrease current consumption and enhance battery lifetime. Wearable systems consist of a biosensor, microcontroller, wireless technology, and a battery for DC supply. The sleep and wake algorithms adopted in previous works used biosensors (e.g., fall detection (FD), heart rate (HR), and temperature) and a microcontroller in wake mode and wireless technology in sleep mode for a period of time during the monitoring of the patient. When the duty cycle ended or any critical health case occurred, the wireless technology woke from sleep mode to transmit patients’ measured data to the EMC.

Some studies [38,39,40,41,42] have proposed a power-saving algorithm for VSMS based on a variable rate sampling strategy. This proposed algorithm focused on controlling the hardware component sampling frequency, allowing a reduction in the proposed devices’ power consumption. Other scholars [16,18] have proposed a technique to decrease the power consumption of the VSMS based on standby and duty-cycle algorithms. Lopez et al. [16] designed a sophisticated health monitoring information system. They monitored various physiological signals, such as the electrocardiogram (ECG), HR, temperature, and tracked patients’ location inside the hospital. The system consisted of a location subsystem, health monitoring subsystem, a wireless sensor network is based on IEEE 802.15.4, and control subsystem. The physiological data were measured by adopting a wearable smart-shirt based on an e-textile. The proposed system was found to have high efficiency with a low transmission rate for data on biometric parameters such as ECG, HR, angle of inclination, activity index, temperature, and battery. The wireless transmission board (WTB) consumed 30, 37, and 20 mA in transmitting, receiving, and standby modes, respectively, while the wearable data-acquisition device (WDAD) consumed about 75 mA, including ECG, HR, accelerometer, and thermometer sensors, A/D convertor, and microprocessor. Thus, the total current consumption of the proposed system (i.e., WTB and WDAD) is about 160 mA. Therefore, the battery life is 3.75 h in traditional operation when a 600 mA/h battery capacity is assumed. However, the adopted power reduction method (i.e., standby mode) decreased the average current consumption to 66.66 and 25 mA for the WDAD and WTB, respectively. Thus, the battery life was prolonged to 9 h for WDAD and 2 days for WTB using the same battery capacity.

Benocci et al. [18] presented an FD system used for monitoring patients based on ZigBee, with supported alarms to maximize system battery life. The system could determine many fall types by using an accelerometer (ACC) sensor attached to the patient’s waist. The authors adopted a duty-cycle algorithm to reduce the current consumed by the proposed device. The suggested system included a wireless protocol based on ZigBee, FD algorithm, an alarm date package transmission using a PC or Smartphone, data collection, alarm, and dispatches for sending a message via SMS or email to an assistant. Experimental results showed that the adopted system had high sensitivity and specificity for detection of falls. The adopted duty cycle algorithm reduced the current consumption to 27.85 mA.

Previous studies have introduced a limitation related to power consumption, where the presented solutions of the above studies still faced a challenge in terms of reducing current consumption and prolonging system battery life. Therefore, our current work differed from these previous works by merging power reduction (i.e., DE algorithm) and energy harvesting techniques based on WPT to charge the proposed HVSMS wirelessly without a wired connection. We subsequently designed and implemented a prototype HVSMS with low power consumption and long battery life that can be employed to monitor the FD and HR of a patient or elderly person.

## 3. System Design

The main purpose of this study was to minimize the power consumption of the proposed HVSMS. Selecting low-power components [43] can contribute to limiting the power consumption of the HVSMS along with the DE algorithm and wireless power transfer (WPT). The HVSMS prototype (Figure 1a) was comprised of six main parts: accelerometer (ACC) for sensing bodily accelerations, heartbeat (HB) sensor for measurement of the heart rate (HR) of the patient, GPS module to determine the geolocations of the patient when falling, the GSM module that was used to send an SMS alert to the telephone number of a doctor or automated system in the EMC, a microcontroller based on the Arduino Pro Mini, and WPT. The GSM model transmits the geolocation information to the EMC via the GSM network. The HVSMS was practically implemented and attached to the upper arm of an elderly patient (Figure 1b). The HVSMS was supplied with a 3.7 V/8400 mAh Li-ion rechargeable battery, which could be charged using WPT (based on an XKT-412 module). In addition, to reduce the HVSMS power consumption, additional components (i.e., three transistors and resistors) were added to control the HVSMS component functions. A transistor (TRAN) based on NPN silicon 2N2222 was used to switch the HVSMS components on (Figure 1c) and off (Figure 1d). The 2N2222 transistor acted as a switching transistor to connect/disconnect the ground pin of the ACC and HR sensors, GPS, and GSM from the ground pin of the DC source.

In general, the ACC sensor communicated via the SPI protocol to the microcontroller, whereas the GSM and GPS modules used serial communications to connect with a microcontroller (Figure 1a). The Arduino Pro Mini was determined to be the most suitable microcontroller board for the proposed system because of its small size, low power consumption, and 20 pins including 14 digital and 6 analog pins, enabling connection to all components; further, it has the capability to support different protocols (I2C, SPI, and serial communications) on a single board based on ATmega328P [44]. The microcontroller represented the core of the HVSMS, and it could be used to acquire sensor and GPS data and process them, subsequently sending the information to the GSM module. In addition, the microcontroller can control the components of the system to reduce the total power consumption based on the DE algorithm.

The ADXL345 (triple axis accelerometer) and heart rate (pulse rate) sensors (Sparkfun, Colorado, United States) were adopted as ACC and HB sensors, respectively. The ADXL345 sensor is small, thin, and ultra-low power (approximately 40 µA in measurement mode). It supports acceleration in three axes (i.e., *x*-, *y*-, and *z*-axes) with high resolution (13 bit) measurements up to ±16 *g* [45]. In addition, static acceleration of gravity in the tilt-sensing application was measured as well as dynamic acceleration resulting from shock or motion. The ACC sensor was supplied with 3.3 V DC voltage by the Arduino Pro Mini. The ACC consumed 0.15 mA of current (based on real measurement). In this paper, the HB sensor was an open-source heart rate monitor that uses a PPG method to observe the heart rate non-invasively. PPG sensors are generally used and worn on the thumb because this method achieves high signal amplification compared to other heart rate measurement methods such as electrocardiography (ECG). In addition, PPG sensors are cost-effective and need less maintenance compared to ECG devices [6]. In the HB sensor, the physical PPG is converted into electrical signals. It basically merges an optical HB sensor with amplification and noise rejection circuitry, making it easy and fast to obtain robust pulse measurements. The HB sensor consists of a transmitter light based on a green LED, a reflective sensor based on a photodetector, an optical filter, and a power supply. The HB sensor has two sides; the first side has a heart shape and is attached to the thumb, while the other side contains most of the components. The HB sensor has three pins; a VCC pin connected to the Arduino +3.3 V, a GND pin connected to the GND pin of the Arduino, and the outputting pin wired to the analog pin (A0) of the Arduino for heart rate measurements. The HB sensor can recognize heart rate by measuring changes in the intensity of the light passing through capillary blood vessels based on the light absorption. The HB sensor measures the heart rate in real time, and beats per minute is computed based on specific algorithm implemented in Arduino Pro mini.

The GSM and GPS modules were chosen to be compatible with the Arduino board. Regarding the GSM module, a SIM800L board was adopted in this work, requiring 3–5 V DC supply voltage and consuming 350 mA of current during transmission of data. It has 88 pin pads of Land Grid Array (LGA) packaging, and all hardware interfaces are provided [46]. Concerning the GPS module, a NEO-M8N board was used. It has high positional accuracy, providing high sensitivity and a short acquisition time while maintaining a low-power system with a maximum current of 70 mA, and outputs the NMEA protocol [47]. Power consumption is a critical issue because health monitoring devices such as the proposed HVSMS are required to operate all day [48]. Figure 2 illustrates all components of the HVSMS with WPT.

## 4. Data-Event Algorithm

One of the most-used methods used for increasing power savings in WBSN applications has been sleep and wake algorithms [49]. Previous works adopting these algorithms have successfully decreased the power consumption of the proposed devices used in health monitoring systems [17,18,19,20,21]. However, in this study, a different approach relative to previous works was practically implemented by proposing an ON/OFF scheme-based event, termed a data-event (DE) algorithm running on a microcontroller such as an Arduino Pro Mini. This algorithm aimed to reduce HVSMS power consumption. Generally, the DE algorithm brings the main power-hungry components of the HVSMS (i.e., GPS and GSM) into a deep sleep (OFF state) until a patient fall event occurs. When a fall happens, the DE algorithm wakes these two components (turn on) to send an alert message within a short time to the EMC, and then returns them to deep sleep until another event occurs. As a result, the power consumption is reduced significantly. The operation of the DE algorithm is described in detail in the following steps:The proposed DE algorithm begins working when the HVSMS is turned on. The initial microcontroller setup is all input/output, turning off the four components: GPS module, GSM module, ACC, and HB sensor, and placing them in sleep mode.The microcontroller sends a control signal via TRAN1 to switch the two biosensors (ACC and HB) from sleep mode to wake mode.In this case, the accelerometer signal (for patient fall detection) and heart rate normal/abnormal can be measured.When a patient fall occurs, the microcontroller checks whether the fall detection threshold (FDT) and fall event time (FET) exceeded the setting values (see Table 1). If the microcontroller measures the acceleration magnitude (*AM*) during the fall, which can be obtained from Equation (1) [50], as being less than or equal to 0.5 *g* with a fall event time of more than 45 ms, the microcontroller issues a decision that the patient is falling regardless of the orientation of the fall. The values of FDT and FET for the ACC sensor were selected to be within the range of standard values as presented in Reference [51], and based on several experiments to ensure high fall detection accuracy, as shown in Table 1.(1)$$AM=\sqrt{A_{x}^{2}+A_{y}^{2}+A_{z}^{2}}$$
where *A_x_*, *A_y_*, and *A_z_* represent the acceleration values in the direction of the *x-*, *y-*, and *z*-axes. The microcontroller then sends two control signals; the first control signal brings the ACC and HB sensor into a sleep mode (OFF state) via TRAN1, and the second control signal awakes the GPS (via TRAN2) and GSM (via TRAN3) modules. The ACC and HB sensors work (awake) most of the time and enter sleep for a fraction of the time (as shown in Figure 3a), on the basis that the patient falls one time per day [10]. In this case, we assume that the fall time (sleep time of ACC and HB sensors) is 2 min and the patient’s monitoring time (awake time of ACC and HB sensors) is 1438 min (1440 − 2).The proposed HVSMS can be used for monitoring patients who have a type of irregular HR called atrial fibrillation, which causes elderly falling. Therefore, it is necessary to measure the HR of the elderly along with fall detection, as presented previously in step 3. These measurements (i.e., fall detection and heart rate) are stored in a specific variable in the microcontroller to be sent to the EMC via the GSM network.Next, the microcontroller achieves two tasks; the first task that the microcontroller energizes the GPS to acquire the geolocation (latitude and longitude) of the elderly fall location based on one or several attempts until the geolocation is acquired. The GPS consumed 80 s to turn on, obtain the geolocation of the patient, and turn off. The second task sends four types of information to the GSM module, including (i) patient name and ID, (ii) accelerometer measurement (i.e., patient fall detection), (iii) status of the heart rate of the patient (normally 60–100 bpm, abnormal if it is out of this range), and (iv) geolocation of the patient. The GSM in turn transmits this information to the EMC through the GSM network by sending an SMS message. However, to ensure that the SMS is delivered to the EMC, the GSM transmits the SMS to the EMC three times, with a delay time of 10 s between each message.When all information has been transmitted, the microcontroller brings the GPS and GSM into a deep sleep (OFF state). The process from steps 5 through 7 consumes 120 s (80 s is required to initiate and establish connections with satellites to acquire the geolocation information and turn off the GPS, whereas 40 s is necessary for turning on the GSM and sending information, then turning off the GSM). Therefore, we assumed that the active time for GPS and GSM to obtain geolocation and information transmission was 2 min (Figure 3b). The active time (i.e., 2 min) represented the maximum time required to obtain the geolocation of an elderly fall using GPS and send it to the EMC through the GSM module. In cases where the GPS signal is not available, the microcontroller sends a control signal to the GPS until the geolocation is acquired, as shown in Figure 4. However, in the current work, the geographic information of the adopted GPS module was experimentally tested and validated for 17 locations in three different cities of Iraq (i.e., Baghdad, Mosul, and Erbil) to investigate the geolocation error of the HVSMS, as presented in our previous paper [47]. The results in Reference [47] showed that the geolocation of the GPS based on NEO-M8N was more reliable and accurate, with a mean absolute error of 1.08 × 10^−5^° and 2.01 × 10^−5^° for latitude and longitude, respectively [47]. During the 17 experiments, we noted that the latitude and longitude locations were obtained safely and no signal losses were recorded in these locations. In addition, in this paper, the proposed fall detection system was designed to work in outdoor environments. The GPS is efficient in outdoor environments and not useful indoors, due to the lack of line-of-sight between the GPS and satellite. Therefore, there is no concern about receiving the signal in outdoor environments based on the adopted fall detection application.Finally, if the microcontroller finds the patient in the normal case (i.e., no fall is happening), the sequence is repeated from step 3. A flow chart of the DE algorithm is presented in Figure 4.

## 5. Power Consumption Model

The lifespan of electronic components such as bio-sensors and wireless technology includes the time elapsed from first transmission until these components lose their working capability. The lifespan of these components relies on the current consumed by each part of the proposed HVSMS. The power consumption of the sensor nodes depends on the number of components. In this paper, the HVSMS consisted of four main components: ACC sensor, HB sensor, GPS module, and GSM module. In addition, the Arduino Pro Mini based on ATmega328P was adopted for data processing and controlling all of these parts to improve the power consumption of the HVSMS. Among these parts, the GPS and GSM modules consumed the highest current [52]. 

The power consumption modeling of the components of the HVSMS can be divided into two parts. Each part includes three power consumers. The first part includes the microcontroller, ACC, and HB sensors, and the second part comprises the power-hungry components such as the GPS and GSM as well as the microcontroller, and thus consume the highest amount of power in this part. The mathematical power consumption models can be explained as follows.

### 5.1. Power Consumption Model of Sensors

As explained above, the power consumption in this part represents the microcontroller, ACC, and HB sensors in traditional operation, as in Equation (2).
(2)$$I_{sensing}=I_{ACC}+I_{HB}+I_{µC}$$
where *I_sensing_* represents the current consumed by ACC and HB sensors and the microcontroller of the HVSMS in traditional operation. *I_µC_* represents the current consumption of the microcontroller based on ATmega328P during measurement. *I_ACC_* and *I_HB_* are the current consumed by the ACC and HB sensors, respectively.

When the DE algorithm is applied, the ACC and HB sensors are in active mode, and they measure the fall detection and heart rate of the patient. However, when the patient falls, both sensors go into deep sleep (OFF state). In this case, the average current consumption of these sensors as well as the microcontroller based on the DE algorithm can be expressed as in Equation (3).
(3)$$I_{sensing\_DE\_Avg}=\frac{T_{Bfall}}{T_{total1}}\;I_{ACC\_Bfall}+\frac{T_{Afall}}{T_{total1}}\;I_{ACC\_Afall}+\frac{T_{Bfall}}{T_{total1}}\;I_{HB\_Bfall}+\frac{T_{Afall}}{T_{total1}}\;I_{HB\_Afall}+I_{µC}$$
where *I_sensing_DE_Avg_* represents the average current consumption based on the DE algorithm. $I_{ACC_{Bfall}}$, $I_{ACC_{Afall}}$, $I_{HB_{Bfall}}$, and, $I_{HB_{Afall}}$ are the current consumption of the ACC and HB sensors in the ON state (before patient fall) and OFF state (after patient fall), respectively. *T_Bfall_* and *T_Afall_*, are the time in ON and OFF states, respectively, and the *T_total1_* is the time elapsed for the next patient or elderly fall. The duty cycle (*DC*_1_) of the patient fall can provide an effective approach for achieving energy efficiency. *DC_1_* can contribute to saving more energy when it has a small value and vice versa, and it can be computed by dividing on-time by the total time (*T_Bfall_/T_total1_*).

Given that *T_Afall_ = T_total1_ − T_Bfall_*, when substituting *DC_1_* in Equation (3), Equation (4) is yielded.
(4)$$I_{sensing\_DE\_Avg}=DC_{1}\;I_{ACC\_Bfall}+\left(1-DC_{1}\right)\;I_{ACC\_Afall}+DC_{1}\;I_{HB\_Bfall}+\left(1-DC_{1}\right)\;I_{HB\_Afall}+I_{µC}$$

In the ON state, the ACC and HB sensors consume a specific amount of power, whereas both sensors go into a deep sleep in the OFF state when the patient is falling. In this case, $I_{ACC_{Afall}}$ and $I_{HB_{Afall}}$ consume zero power. Consequently, Equation (4) can be shortened to Equation (5).
(5)$$I_{sensing\_DE\_Avg}=DC_{1}\;I_{ACC\_Bfall}+DC_{1}\;I_{HB\_Bfall}+I_{µC}$$

### 5.2. Power Consumption of Wireless Technology

The power consumption in this part represents the microcontroller and GPS and GSM modules in traditional operation, as in Equation (6).
(6)$$I_{wireless}=I_{GSM}+I_{GPS}$$
where *I_wireless_* is the current consumed by the GSM, GPS, and microcontroller of the HVSMS in traditional operation. *I_µC_* represents the current consumption of the microcontroller based on the ATmega328 during data reception from the GPS and data transmission to the GSM. *I_GPS_* and *I_GSM_* are the current consumed by GPS and GSM modules, respectively.

When the DE algorithm is considered, the GSM and GPS modules are in a deep sleep mode in a normal patient case. In contrast, when the patient falls, the GSM and GPS wake up to handle their data. The GPS sends the geolocation to the microcontroller, and the GSM sends information that includes patient name and ID, patient fall detection, geolocation of fall incident, and the patient’s heart rate status to the EMC. Accordingly, GSM and GPS consume a significant amount of power. In this case, the average current consumption of these components and the microcontroller based on the DE algorithm can be expressed as in Equation (7).
(7)$$I_{wireless\_DE\_Avg}=\frac{T_{Bfall}}{T_{total2}}\;I_{GPS\_Bfall}+\frac{T_{Afall}}{T_{total2}}\;I_{GPS\_Afall}+\frac{T_{Bfall}}{T_{total2}}\;I_{GSM\_Bfall}+\frac{T_{Afall}}{T_{total2}}\;I_{GSM\_Afall}$$
where *I_wireless_DE_Avg_* is the average current consumption based on the DE algorithm. $I_{GPS_{Bfall}}$, $I_{GPS_{Afall}}$, $I_{GSM_{Bfall}}$, and $I_{GSM_{Afall}}$ are the current consumption of the GPS and GSM in the OFF state (before patient fall) and ON state (after patient fall), respectively. *T_Bfall_* and *T_Afall_* are the time in the OFF state and ON state, respectively, and the *T_total2_* is the time elapsed for the next patient or elderly fall. In a similar manner as presented above, *DC_2_* can be computed by dividing ON time by the total time (*T_Afall_/T_total1_*).

Given that *T_Bfall_ = T_total1_ −T_Afall_*, when substituting *DC_2_* into Equation (7), Equation (8) is obtained.
(8)$$I_{wireless\_DE\_Avg}=(1-DC_{2})\;I_{GPS\_Bfall}+DC_{2}\;I_{GPS\_Afall}+(1-DC_{2})\;I_{HB\_Bfall}+DC_{2}\;I_{HB\_Afall}$$

In the ON state (patient fall), the GPS and GSM modules consume a significant amount of power, whereas both modules enter a deep sleep in the OFF state when the patient is in a normal situation. In this case, $I_{GPS_{Bfall}}$ and $I_{GSM_{Bfall}}$ consume zero power. Consequently, Equation (8) can be reduced to Equation (9).
(9)$$I_{wireless\_DE\_Avg}=DC_{2}\;I_{GPS\_Afall}+DC_{2}\;I_{HB\_Afall}$$

The total current consumption of the HVSMS in traditional operation can be computed by adding Equations (2)–(6). However, Equations (5) and (9) can be combined to produce the average current consumption of the HVSMS based on the DE algorithm. Equations (10)–(13) are the total current consumption of the HVSMS in traditional operation and using the DE algorithm.
(10)$$I_{total_{traditional}}=I_{sensing}+I_{wireless}$$
(11)$$I_{total_{traditional}}=I_{ACC}+I_{HB}+I_{GSM}+I_{GPS}+I_{µC}$$
(12)$$\;I_{total\_DE\;\_Avg}=I_{sensing\_DE\_Avg}+I_{wireless\_DE\_Avg}$$
(13)$$I_{total\_DE\;\_Avg}=DC_{1}\;I_{ACC\_Bfall}+DC_{1}\;I_{HB\_Bfall}+DC_{2}\;I_{GPS\_Afall}+DC_{2}\;I_{HB\_Afall}+I_{µC}$$

The average current consumptions presented in Equations (11) and (13) can be used to determine the power saving of the HVSMS in traditional and DE algorithm operation, respectively, as in Equation (14). In addition, the battery lifetime (*Battery_LT_*) of the HVSMS can be calculated based on Equation (15) [53].
(14)$$Power\;saving\;\left(\%\right)=\;\left(1-\;\frac{I_{total\_DE\_Avg}}{I_{total\_traditional}}\right)\;\times \;100$$
(15)$$Battery_{LT}=\;\frac{C_{battery}\;}{I_{total}}\times \left(1-discharge\;safety\right)$$
where *C_battery_* is the initial battery capacity, which used as a DC power supply, where two Li-ion batteries (3.7 V/8400 mAh) were adopted in this work. *I_total_* is the total current consumption of the HVSMS and represents the total current in Equations (11) and (13) for traditional operation and total average current consumption based on the DE algorithm, respectively. *Discharge safety* is the percentage of the considered battery capacity that is never employed, e.g., if a Li-ion battery is employed to run the HVSMS, we must never discharge it below 20% or it may be damaged. Therefore, in this work, the estimated battery life adopted a discharge safety of 20%. 

### 5.3. WPT Charging Model

Wireless power transfer (WPT) has recently become common for charging portable devices [54,55]. In the current study, this approach was considered for charging the batteries of the HVSMS. The WPT technology can be used to charge the batteries of the HVSMS wirelessly. In this study, an XKT-412 module [56] was adopted for WPT, as shown in the schematic diagram in Figure 5. This module consists of transmitter and receiver circuits. The transmitter circuit contains a voltage regulator and oscillator circuit and includes a transmitter coil (*Tx*) and parallel capacitor (C_1_). The transmitter circuit generates approximately 200 kHz frequency on the basis of the resonance frequency in Equation (16). The receiver circuit comprises a receiver coil (*Rx*) and parallel capacitor (C_2_), AC/DC converter diode, and a voltage regulator with input and output capacitors C_3_ and C_4_, respectively, as shown in Figure 5, to filter the DC signal from unwanted noise. The adopted WPT module shown in Figure 6 is characterized by a stable power supply, magnetic resonator coupling (MRC) technique, insulation coils, 90% power transfer efficiency, and compact design.
(16)$$f_{o}=\frac{1}{2\pi \sqrt{L_{Tx}C_{1}}}$$
where *f_o_* is the resonant operating frequency, *L_Tx_* is the transmitter coil, and *C_1_* is the parallel capacitor of the resonant circuit. 

In this paper, two Li-ion batteries were used to supply the HVSMS with a voltage in the range of 3.7 to 4.2 V. The DC output voltage and current of XKT-412 were 5 V/1 A—this voltage value is higher than the voltage of the HVSMS battery. Therefore, a DC-to-DC step-down voltage converter was used to convert the voltage down to be consistent with the charging voltage of the HVSMS battery, as shown in the hardware diagram of the WPT, presented in Figure 6. The TP4056 board was used as a DC–DC converter [57] to convert the output voltage of the WPT (XKT-412) to 4 V. A simple experiment was conducted to measure the output voltage of the WPT module at different distances. The distance between transmitter and receiver coils was tested from 0 to 5.5 cm, with a distance step of 0.5 cm, until 5.5 cm was reached. The results of this experiment are highlighted in Section 6. 

## 6. Results and Discussions

The results of the HVSMS power consumption are presented and discussed in the following subsection.

### 6.1. Current Consumption Measurements

The HVSMS was practically implemented to measure and detect the heart rate and the fall of the patient. The current consumption of each component in the HVSMS was measured using a digital multimeter device (UNI-T/UT33D), as shown in the testbed in Figure 7. The current consumption of each component of the HVSMS was measured before and after a patient fall, as well as time consumption, as shown in Table 2. In addition, the average current based on the DE algorithm was calculated for HB and ACC sensors in the sensing and processing phase (Equation (5)), and for GPS and GSM for information transmission phase (Equation (9)). Moreover, the total and average current consumption of the HVSMS were estimated before (i.e., traditional operation without any power reduction technique) and after applying the DE algorithm based on Equations (11) and (13), respectively. Consequently, power savings and battery life were estimated. The measurement results presented in Table 2 indicate that a significant improvement in power consumption was achieved. 

However, the table discloses that the GPS and GSM modules consumed a high amount of power (25.1 and 51.5 mA, respectively) relative to other components in the HVSMS. It is worth mentioning that in this paper, the components of the HVSMS, which include the ACC, HB, GSM, and GPS modules, can go into sleep mode to save energy. However, the microcontroller cannot go into sleep mode and always stays working, as shown in Table 2, because it continuously monitors the heart rate and detects falls of the patient, acquires the geolocation from the GPS, and sends information to the GSM, as well as controlling the ON/OFF state of the GSM and GPS modules and the ACC and HB sensors. To evaluate the performance of the HVSMS in terms of power consumption, the power savings and battery life span are discussed in Section 5.2.

### 6.2. Current Consumption Based on DE Algorithm

Once the DE algorithm was applied to the HVSMS, the power consumption was significantly reduced. Figure 8 illustrates the measurements of current consumption of each HVSMS component before (i.e., traditional operation) and after applying the DE algorithm. Based on the DE algorithm, the average current consumption for each component was developed to 0.149, 2.195, 0.034, and 0.071 mA. Consequently, the overall average current consumption was reduced to 9.35 mA relative to traditional operation (i.e., 85.85 mA). Figure 8 reveals that the GSM and GPS came in first and second, respectively, for consuming power in traditional operation. However, when the DE algorithm was applied, the power consumption was considerably reduced. This reduction in power consumption can be attributed to the GPS and GSM being in the OFF state most of time (before patient fall) and in the ON state for only a fraction of time (after patient fall occurs to transmit their information). On the other hand, there was a slight difference in power consumption in ON and OFF states of the HB and ACC sensor because they were in the ON state (before patient fall to measure and monitor the heart rate and fall) most of the time and in the OFF state (after patient fall) for only a fraction of time.

### 6.3. Battery Lifetime Estimation and Power Saving based on DE Algorithm

The application of the DE algorithm verified that this approach can reduce the current consumption of the proposed HVSMS to 9.35 mA. Consequently, battery life can be extended to 30 days (estimated based on Equation (15)) until charging is needed again based on WPT. In this study, two 3.7 V/4200 mAh Li-ion batteries were connected in parallel to obtain a battery capacity of 3.7 V/8400 mAh to increase the HVSMS lifespan, where the battery life of the proposed system was four days under traditional operating conditions. Table 3 supports the results presented in Figure 8, where the battery life of the HVSMS can be seen to have been prolonged. Accordingly, a power saving of 89% in the HVSMS was achieved (estimated based on Equation (14)). Table 3 shows that the proposed DE algorithm significantly improved the power consumption of the HVSMS relative to traditional operation. However, when the battery capacity increased, the lifetime of the HVSMS also increased and vice versa. Clearly, the proposed algorithm succeeded in terms of decreasing HVSMS current consumption. The results show that the proposed HVSMS worked with substantially lower power consumption using power saving measures.

## 7. Performance Evaluation of Wireless Power Transfer

The WPT technology adopted in this work was used to charge the 3.7 V/8400 mA Li-ion battery of the HVSMS. This battery requires at least 4 V to be charged. WPT based on the XKT-412 module was able to supply 5 V/1 A. Therefore, a DC-to-DC voltage down converter was used to adapt the output voltage of the module (i.e., 5 V) to 4 V. To evaluate the performance of the adopted WPT, the output voltage was investigated with different distances (0–5.5 cm) to explore the best charging distance, as shown in Figure 9. The figure illustrates the results of the output DC voltage of the WPT with different distances between transmitter and receiver coils. The DC output voltage decreased with increasing distance between the transmitter and receiver coils and vice versa. The results show that the WPT technology was able to charge the HVSMS battery up to a distance of 1.5 cm between transmitter and receiver coils, which can supply 4 V with 1 A. Based on these results of output current and voltage, the HVSMS battery can be safely charged within 10 h. The combination of the DE algorithm and WPT allows the HVSMS to be charged once for 10 h (charge time) every 30 days, whereas, for the same charge time, the WPT charges the battery of HVSMS once every 4 days without applying the DE algorithm (i.e., traditional operation). Charging the battery of the HVSMS based on the WPT technique presents some gains, such as dispensing with the physical connection for battery charging, and the patient can move freely without the restrictions of electrical wires. In contrast, wired charging methods present some shortcomings; for example, there are electrical losses during plug-out and plug-in with charging devices, and the environmental conditions (i.e., humidity and temperature) can reduce the efficiency and reliability of charging devices.

## 8. Power Consumption Comparison with Related Works

The power consumption of the proposed HVSMS using a DE algorithm was compared with the current drain of other systems outlined in existing studies [16,17,18,19,20,21,24,37,38,39,40,41,42] on patient vital signs monitoring systems (Figure 10) to confirm the proposed HVSMS. The performance of the current work in terms of current drain was accomplished by relying on the designed system in Section 3 and was confirmed by the implementation prototype (Section 3) and proposed DE algorithm in Section 4, and current consumption modeling (Section 5) and current measurements (Section 6.1). The proposed DE algorithm was compared with the existing related articles on the basis of the values documented during measurements in the previous studies. The performance of the previous techniques is presented in Section 2 in terms of current consumption.

The current consumption of the proposed HVSMS was minimized based on the DE algorithm to 9.35 mA using DC1 (for HB and ACC sensors) and low DC2 (for GPS and GSM modules) relative to traditional operation (85.85 mA). The results revealed that the current study method outperformed other approaches in related works in terms of current consumption, as shown in Figure 10.

## 9. Conclusions

A health monitoring system was practically designed and implemented in this paper, namely an HVSMS. The proposed system was used for monitoring the acceleration and HR measurement of a patient, and then used to send a text message to the EMC to alert them when a patient fall or any other event has occurred. The HVSMS consists of a microcontroller, GPS module, GSM module, ACC sensor, and HB sensor; these components are assembled in the power electronic board and connect to two Li-ion batteries (3.7 V/8400 mAh). The upper arm of the subject’s body was selected as a location to attach the proposed system. Merging between the DE algorithm and WPT was achieved in this paper. Power consumption of the HVSMS was reduced based on the DE algorithm. The DE algorithm is uncomplicated and does not require significant time for data computation and processing and making a decision. However, the DE algorithm considerably improved the power consumption of the HVSMS. Consequently, the battery life was extended and power savings was developed for the HVSMS. The main principle of the DE algorithm was to ensure that most components consumed low current in sleep mode until any events occurred. Wireless technology such as GSM and GPS modules consumed most of the current, and therefore these parts were designed in this work to function only when the patient falls, and to return to sleep mode and to turn off when the mission is complete. 

The method outlined in this study decreased the current consumption of the HVSMS from 85.85 mA during traditional operation to 9.35 mA by using the DE algorithm, which improved the battery life of the proposed system by allowing the device to work for 30 days without charging the battery, whereas it only worked for 4 days during traditional continuous operation. In addition, the proposed algorithm was successful because it was able to reduce by 89% the power consumed by the HVSMS, and the WPT technology was adopted to charge the HVSMS battery wirelessly once every 30 days for 10 h. The power consumption could be further reduced in future work by (i) adopting a standalone microcontroller, which would reduce the power consumption of the microcontroller in active mode from 6.9 to 0.5 mA at working frequency of 1 MHz [58], thereby significantly improving current consumption and minimizing the size of the system; (ii) assuming that the elderly person falls once every two days or more; (iii) reducing the current consumption of the HB and ACC sensors before a fall by adopting an additional duty cycle in this period, leading to the current consumption being further reduced to half or less; and (iv) adopting a specific intelligent technique or machine learning method. In addition, in the future the size of the HVSMS can be further reduced by using small surface mount technology (SMT) chips instead of the modules and by assembling the device on one printed circuit board (PCB) in order to make it marketable and affordable.

## Figures and Tables

**Figure 1 sensors-19-04452-f001:**
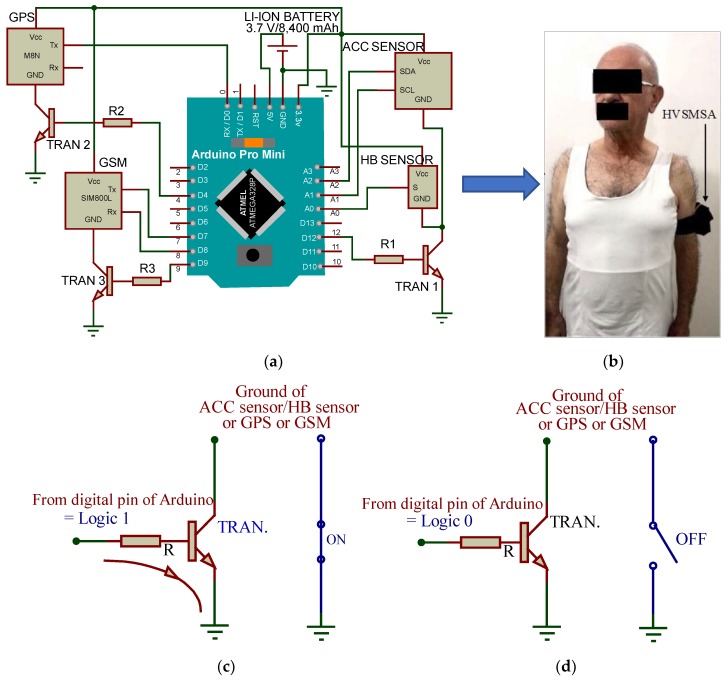
Human vital signs monitoring system (HVSMS) with (**a**) schematic diagram and (**b**) attached to upper arm of elderly patient, (**c**) switching transistor at on state, and (**d**) switching transistor at off state.

**Figure 2 sensors-19-04452-f002:**
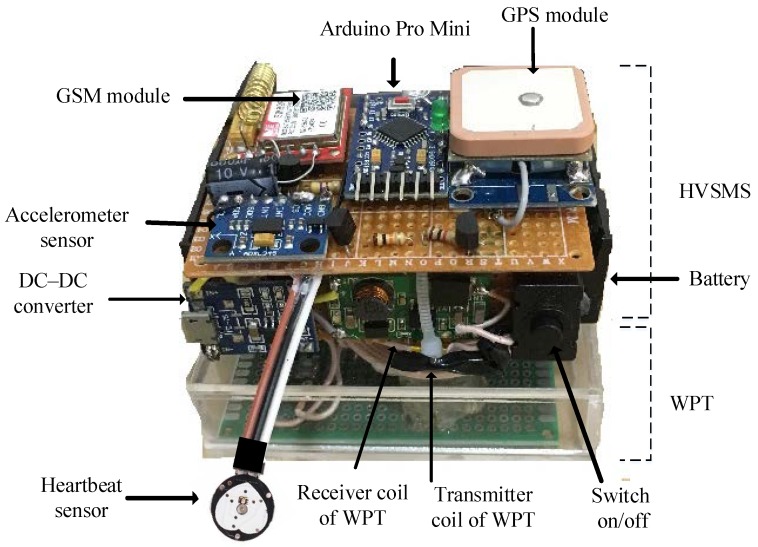
Hardware of entire HVSMS setup.

**Figure 3 sensors-19-04452-f003:**
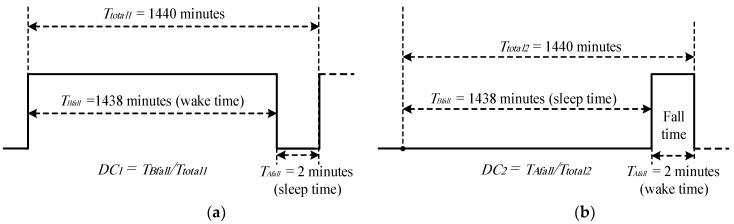
Timing diagram of (**a**) ACC and HB sensors (monitoring time) and (**b**) GPS and GSM modules (fall time).

**Figure 4 sensors-19-04452-f004:**
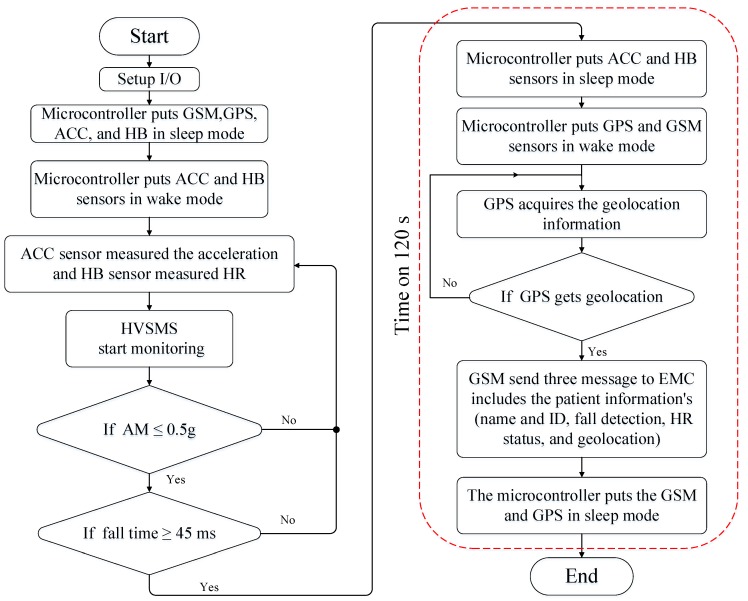
Flow chart of data-event (DE) algorithm.

**Figure 5 sensors-19-04452-f005:**
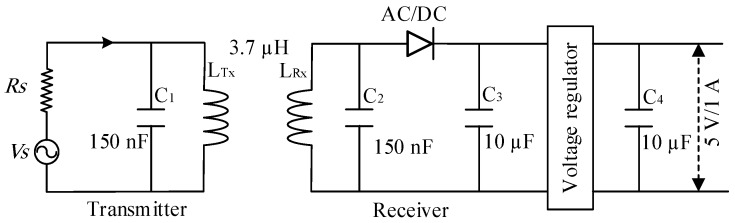
Schematic diagram of XKT-412 module wireless power transfer.

**Figure 6 sensors-19-04452-f006:**
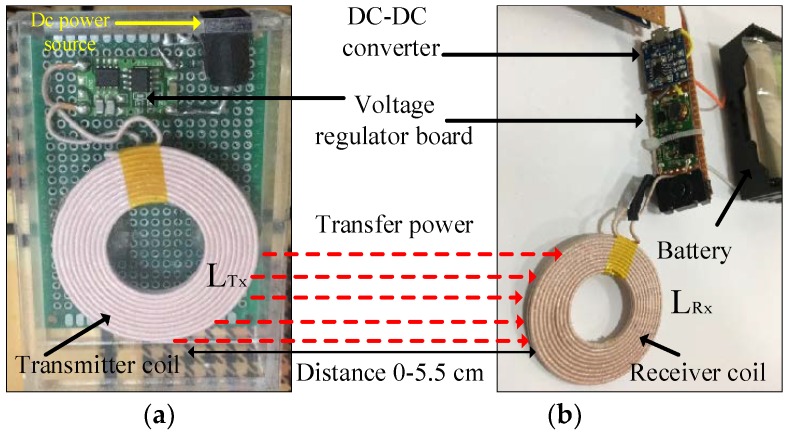
Wireless power transfer with (**a**) transmitter circuit and (**b**) receiver circuit with an FC-75 board.

**Figure 7 sensors-19-04452-f007:**
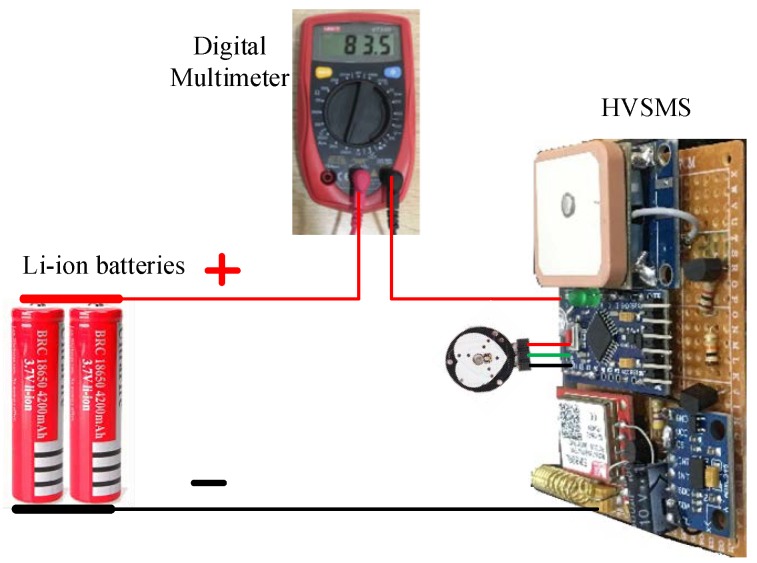
Current consumption measurements of the HVSMS when patient falls.

**Figure 8 sensors-19-04452-f008:**
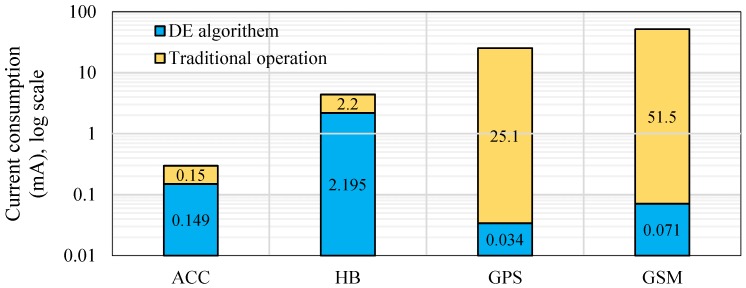
Power consumption of each component in HVSMS before and after applying the DE algorithm.

**Figure 9 sensors-19-04452-f009:**
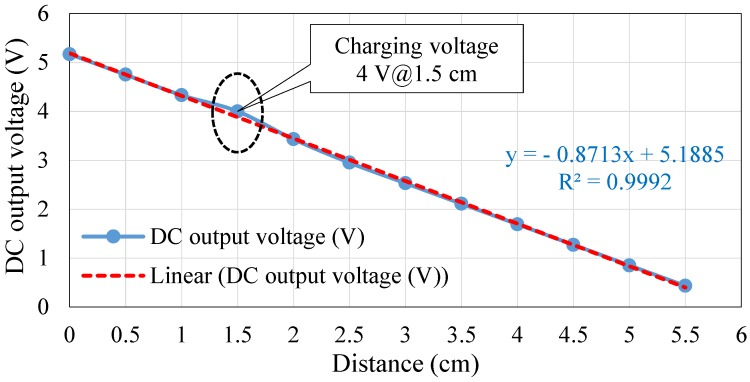
Relationship between distance and DC output voltage of the WPT.

**Figure 10 sensors-19-04452-f010:**
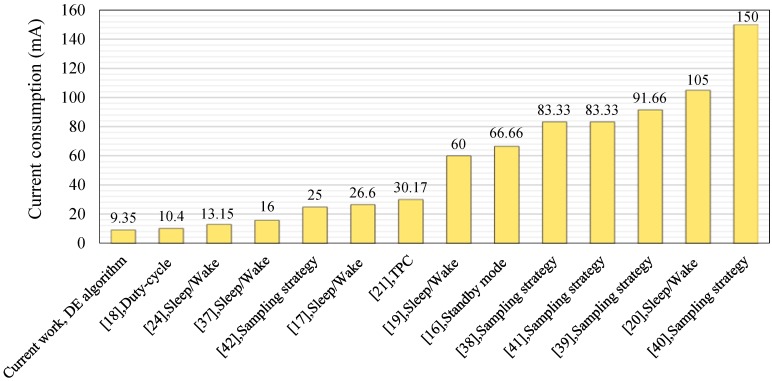
Current consumption comparison of HVSMS with previous works.

**Table 1 sensors-19-04452-t001:** Standard and adopted threshold values of the ACC sensor.

Parameters	Standard Values [51]	Selected ACC Threshold
FDT	0.313–0.563 g	0.5 g
FET	20–70 ms	45 ms

**Table 2 sensors-19-04452-t002:** Measurement of current consumption and time profile of HVSMS before and after applying the DE algorithm.

Parameter	HB Sensor	ACC Sensor	Arduino Pro Mini	GPS	GSM
*I* (mA)	2.2 ^$^	0.15 ^$^	6.9 *^,&^	25.1 ^$^	51.5 ^&^
*I* _DE_Avg_	2.195	0.149	6.9	0.034	0.071
*T_Bfall_* (minutes)	1438	1438	always on	1438	1438
*T_Afall_* (minutes)	2	2	always on	2	2
*DC_1_*	0.998 (1438/1440)		1	---	
*DC_2_*	---		1	0.00138 (2/1440)	
*I_sensing_DE_Avg_* = 9.2453 mA			Equation (5)		
*I_wireless_DE_Avg_* = 0.1057 mA			Equation (9)		
*I_total_traditional_* = 85.85 mA			Equation (11)		
*I_total_DE_Avg_* = 9.35 mA			Equation (13)		
Power savings = 89%			Equation (14)		
Battery_LT_ in traditional operation= 97 h (4 days)			Equation (15)		
Battery_LT_ based on DE algorithm = 718.7 h (30 days)			Equation (15)		

^$^ practically measured at 3.3 V, ^&^ practically measured at 3.7 V, * without power LED.

**Table 3 sensors-19-04452-t003:** Battery life and power savings of HVSMS based on the DE algorithm and traditional operation at 8400 mAh battery capacity.

Parameter	DE Algorithm	Traditional Operation
Current consumption (mA)	9.35	85.85
Battery life time (h)	718.7 (30 days)	97 (4 days)
Power savings (%)	89	------

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
