# Peer review of "Energy-Efficient Elderly Fall Detection System Based on Power Reduction and Wireless Power Transfer"

_sensors, 2019, doi:10.3390/s19204452_

Round 1

Reviewer 1 Report

In this paper authors propose a human vital sign monitoring system (HVSMS). The goal is to measure vital parameters of the elderly. Besides to this goal, authors intend to optimize the energy consumption of a portable device carried by elderly persons. The paper is well structure however there is a lack of innovation. It seems just a description of a system (engineering) and doesn’t describe relevant things to improve the state-of-the-art.
Questions:
- How the person physiognomy affects the correct functionality of the system?
- How the age influence can be considered?
- How the skin conductivity/skin coupling is considered?
A calibration or parameterization model is missing.
The DE algorithm (that is not described in detail) may fails in several situations, for example what happens when GPS signal is not available, or a long fixe time is necessary?
what means the time referred in fig. 4? (120 seconds in maximum or average)?
What happens if no GSM connections exists or the SMS is not delivered? Is there any confirmation from SMS receiver’s? Is there any retry mechanism implemented to keep a safety behavior of the patient (elderly person)? (no fall)
In my opinion this manuscript just describes a system in a generic way and doesn’t describe the innovation details that are needed to develop/present a reliable solution to follow/monitor elderly persons.

Author Response

Reply to the reviewer 1 as in attached file

Reviewer 2 Report

Abstract

A battery life of 37 days is indicated, which is calculated assuming the full battery charge of a 8400 mAh cell (mentioned later) can be used, which is not correct. The circuit will fail long before the battery is fully discharged.

1. Introduction

no remarks

2. Related works

"The adopted standby mode algorithm decreased the current consumption to 244.5 mA during the transmission process"

Normally the average current would be mentioned here, as the peak current occurring during the transmission process in normally being drawn when the system is *not* in standby mode.

3. System design

The ADXL345 accelerometer is said to consume only 40 microAmps in measurement mode. According the Analog Device's datasheet this is only the case for VS=2.5V and VDD_IO=1.8V. The voltages are assumed to be 5V in your design, according to Fig. 5.

Fig.1: Circuit diagram

The power of the peripheral components is "switched off" using a NPN transistor in the GND connection. Normally you would switch off the VCC using a PNP transistor in the positive supply; after setting all the IO lines low.

Did the circuit as proposed actually succeed in switching the peripheral IC's on and off properly?

4. Data-event algorithm

Steps 1 to 8

I do not understand step 4.

What is meant by a fall state time of 2 minutes and a normal state time of 1438 minutes?

In step 6, how long does it take the GPS unit to determine its position after power on?  I see this is explained in step 7 whereas it would better be mentioned in step 6.

Do you actually detect the acceleration during the fall, or merely the change in position after the fall?

Equation (1) determined AM, regardless of orientation, but how is this value further used?  AM <= 0.5 g sets off the alarm state and is the only trigger for this?

5 Power consumption model

6 Results and discussions

6.1 Does the last paragraph this subsection actually say that the designed system cannot go in sleep mode and hence all the values about reduced power consumption are only theoretical, and were never measured?

Are the results in Fig 8 measured or calculated? From the text this is not entirely clear.

The graph in Fig. 9 does not really contribute a lot, as the underlying information constitutes out of only 2 positive real numbers.

7 WPT

8 Power consumption

9 Conclusion.

References.

Final remark:

I missed information about the heartbeat sensor used in this paper.

Author Response

Reply to the reviewer 2 as in attached file

Reviewer 3 Report

The paper is well written, but the core research question is not clear.Please clearly explain what is the problem you are solving.

There numerous fall detection devices in the market. Why do you choose to measure the heart rate in yours. How is this useful?

The power saving algorithm is a simple if-then rule. Did you consider applying machine learning or other more adaptive techniques? This could improve the interest of the readers.

What is the recharging time with and without the algorithm. It is not clear that there is real gain, as fall events occur rarely.

The device looks bulky in the photo. Have you considered ergonomy, user friendliness, design aspects?

It is not clear where the GSM module forwards the data. Clarifications are needed.

Author Response

Reply to the reviewer 3 as in attached file

Round 2

Reviewer 2 Report

Dear author's,

Thank you for responding to my remarks and adapting the paper.

Here are my remarks on the revised text.

Kind regards

Reviewer 2

------------------------

Line 139

***It would be nice to add the original power consumption, and the initial number of days autonomy, so the reader can easily see the improvement.

----------------------------------------------------------

Line 170
***I still disagree on this topic. This is not the right way to switch a sensor. The GND line is the reference for your digital communication and switching it witch a transistor is not good practice. It might work in the lab, but is less reliable than keeping solid GND connections between processor and sensors and switching the positive supply.

----------------------------------------------------------

Line 188

***Please explain the abbreviation PPG in the text. The brand and type of sensor should be mentioned.

-------------------------------------------------------------

Line 474 Figure 9

*** The figure shows two straight lines, with angles depending on the average current consumption of the device. Hence, a graph is used to convey only the information of two constant values. This kind of graph does not fit in a scientific publication. Every reader knows the battery life is proportional to the battery capacity and the current drawn. Please just mention the relevant number in a small table, instead of the graph.

Author Response

Reply to reviewer 2 as in attached file

Reviewer 3 Report

All comments have been successfully addressed in the resubmission.

Author Response

Reply to reviewer 3 as in attached file.
